# Placental DNA methylation changes and the early prediction of autism in full-term newborns

**Ray O. Bahado-Singh**[1], **Sangeetha Vishweswaraiah**[1], **Buket Aydas**[2], **Uppala Radhakrishna**[1]*

**1** Department of Obstetrics and Gynecology, Oakland University William Beaumont School of Medicine, Royal Oak, MI, United States of America, **2** Department of Healthcare Analytics, Meridian Health Plans, Detroit, MI, United States of America

* Uppalar99@gmail.com

**Data Availability Statement:** Data files are available from Chairman, Department of OB & Gyn, Royal Oak, MI email: Ray.Bahado-Singh@beaumont.org on reasonable request.

## Abstract

Autism spectrum disorder (ASD) is associated with abnormal brain development during fetal life. Overall, increasing evidence indicates an important role of epigenetic dysfunction in ASD. The placenta is critical to and produces neurotransmitters that regulate fetal brain development. We hypothesized that placental DNA methylation changes are a feature of the fetal development of the autistic brain and importantly could help to elucidate the early pathogenesis and prediction of these disorders. Genome-wide methylation using placental tissue from the full-term autistic disorder subtype was performed using the Illumina 450K array. The study consisted of 14 cases and 10 control subjects. Significantly epigenetically altered CpG loci (FDR p-value <0.05) in autism were identified. Ingenuity Pathway Analysis (IPA) was further used to identify molecular pathways that were over-represented (epigenetically dysregulated) in autism. Six Artificial Intelligence (AI) algorithms including Deep Learning (DL) to determine the predictive accuracy of CpG markers for autism detection. We identified 9655 CpGs differentially methylated in autism. Among them, 2802 CpGs were inter- or non-genic and 6853 intragenic. The latter involved 4129 genes. AI analysis of differentially methylated loci appeared highly accurate for autism detection. DL yielded an AUC (95% CI) of 1.00 (1.00–1.00) for autism detection using intra- or intergenic markers by themselves or combined. The biological functional enrichment showed, four significant functions that were affected in autism: quantity of synapse, microtubule dynamics, neuritogenesis, and abnormal morphology of neurons. In this preliminary study, significant placental DNA methylation changes. AI had high accuracy for the prediction of subsequent autism development in newborns. Finally, biologically functional relevant gene pathways were identified that may play a significant role in early fetal neurodevelopmental influences on later cognition and social behavior.

**Funding:** BA is a paid employee of Meridian HealthComms Ltd. The funder provided support in the form of salaries for authors [BA], but did not have any additional role in the study design, data collection, and analysis, decision to publish, or preparation of the manuscript.

**Competing interests:** The authors have read the journal's policy and have the following competing interest: BA is a paid employee of Meridian HealthComms Ltd. There are no patents, products in development, or marketed products associated with this research to declare. This does not alter our adherence to PLOS ONE policies on sharing data and materials.

## Introduction

Autism is a neurological disorder that is characterized by social, communication, and cognitive dysfunction and is clinically heterogeneous [1–3]. It generally manifests before the age of 3 years [4]. It has been estimated that about 2.21% of adults are living with some form of autism spectrum disorder (ASD) [5]. Poor health care outcomes including early mortality are documented among the ASD group compared to peers of the same age group [6].

Etiologically, ASD is mediated by genetic, epigenetic, environmental influences, and immune dysfunction [3]. DNA methylation, an epigenetic mechanism, uses cytosine methylation or attachment of a carbon atom to regulate gene expression without altering the DNA nucleotide sequence. DNA methylation changes in autism manifest in the brain, placenta, and blood [7]. The placenta is a transient organ that controls the biological interaction between the maternal and fetal environment [8]. The placenta produces neurotransmitters such as serotonin, dopamine, norepinephrine/ epinephrine along with nutrient transfer and gases that contribute to a major degree to fetal neurodevelopment [9] and neurodevelopmental disorders [8].

Artificial Intelligence (AI) is a branch of computer sciences in which machines, with limited explicit programming, perform tasks that normally require human intelligence. AI applications in the biological sciences from imaging to genomic analysis [10, 11] represent an exciting new development. AI is ideal for the analysis of the large volume of data generated from 'omics' experiments [12–14].

Placental methylome changes are known to occur in the placenta of autistic children [15, 16]. Our prior studies have shown the power of AI and machine learning analysis of omics data for the detection of different phenotypes [17–20]. For example, we have earlier reported DNA methylation changes in newborn leukocyte DNA combined with AI for the detection of autism [13]. Prematurity is associated with altered brain macro and microstructure and is a well-recognized risk factor for autism [21]. Most cases of autism however occur in term infants and the mechanisms are on the surface less intuitive. Based on the above considerations, therefore, we evaluated the utility of placental epigenomics to help elucidate the molecular mechanisms of autism in children born at term. Further, given the recognized benefit of early detection in improving outcomes in autistic infants, we evaluated the utility of AI platforms for the accurate prediction of autism.

## Materials & methods

Study methods were previously described [13] and are summarized herein. Wayne State University, Detroit MI, Royal Oak, MI, USA granted IRB approval. The study group consisted of 14 term autism cases (7 males, 7 females) and 10 term ethnicity-matched normal controls (5 males, 5 females) that were not delivered prematurely. The parent or legal guardian of the child provided written consent. The pediatric medical records were reviewed. We identified children that were diagnosed with classic autism by a pediatric neurologist, and who had delivered at WSU, and for whom placental histology was performed. DNA was extracted from formalin-fixed, paraffin-embedded (FFPE) residual placental tissue that was archived after the histological exam. We used the term singleton birth without a diagnosis of autism or any known or suspected genetic syndromes or birth defects as controls and for whom FFPE placental tissue block was available. These control children had been followed birth in the regular pediatric clinic with no diagnosis of developmental or brain disorders. This was confirmed based on chart review and by the parent/ guardian. Exclusion criteria for cases also included the absence of other known or suspected genetic syndromes or major anomalies. The autism cases were diagnosed before 2014 as per the prevailing classification by the American

Psychiatric Association (DSM-IV). The diagnostic category used was the autistic disorder sub-type. In 2014 the classification system was changed to ASD encompassing the five different subtypes: autistic disorder, pervasive developmental disorder- not otherwise specified, Asperger syndrome, Childhood Disintegrative Disorder, and Rett syndrome. Limited demographic and clinical data were obtained from the delivery records. DNA was extracted from FFPE tissue using Qiagen Gentra Puregene Tissue Kit (Gentra systems MN, USA), and subsequent bisulfite conversion of 500 ng of each extracted DNA using EZ DNA Methylation-Direct Kit (Zymo Research, Orange, CA) was performed according to the respective kit protocols.

### Genome-wide DNA methylation profiling and statistical analysis

The Illumina HumanMethylation450 BeadChip (450K), covering 485,000 CpG sites (Illumina, Inc., California, USA) was used to perform DNA methylation profiling. The samples were randomized and processed together to minimize the batch effect. The processed array chips were fluorescently stained and imaged on Illumina iScan. CpG loci within 10bp of SNPs were excluded because the latter can alter the methylation status of proximate cytosine loci. Statistical and bioinformatic analyses were performed followed by data processing and quality control. A β-value based on the ratio of methylated and unmethylated signal intensities were calculated for each CpG loci using GenomeStudio software as detailed in our prior publications [12, 22]. FDR (Benjamini-Hochberg) p-value < 0.05 was considered significant. R-packages dplyr, reshape2, and ROCR were used to calculate Area Under the Receiver Operating Characteristic (AUC-ROC) curves with 95% CI for ASD prediction. An unsupervised Principal Component Analysis (PCA) and heatmap were generated using an online tool "MetaboAnalyst 4.0" which is based on the R program [23] (Fig 1).

### Prediction of ASD CpG markers using Artificial Intelligence (AI)

Our AI analytic methods have been extensively described previously [13, 20]. Six AI-based platforms were used to predict autism based on CpG methylation. These algorithms were, Deep Learning (DL), Support vector machine (SVM), Generalized Linear Model (GLM), Prediction Analysis for Microarrays (PAM), Random Forest (RF), and Linear Discriminant Analysis (LDA) [24]. Analysis of placental intragenic, extragenic CpG methylation markers and both combined were evaluated in autism prediction. The details were published earlier [20].

### Biological functional enrichment analysis

The genes with significant CpG methylation changes were used for the disease enrichment assessment using Ingenuity Pathway Analysis (IPA) (Qiagen IPA) system. The genes with Entrez identifiers recognized by IPA were mapped. The disease mechanism associated with autism with a statistical significance of p-value <0.05 was considered.

### Results

The study cases and controls are not significantly different in terms of maternal age, race, and gestational age (S1 Table). A total of 9655 CpGs were found to have significantly altered methylation in the full-term birth autism cases. Among them, 2802 CpGs were intergenic markers and 6853 (4129 genes) were intragenic CpG. These involved the Transcription Start Site (TSS) 200, TSS1500, 5' UTR, 1st exon, gene body, and 3' UTR. PCA showed a clear separation of ASD cases and normal control subjects (S1 Fig). The hierarchical clustering showed separation of CpG markers based on hyper and hypomethylation status, depicted in S2 Fig.

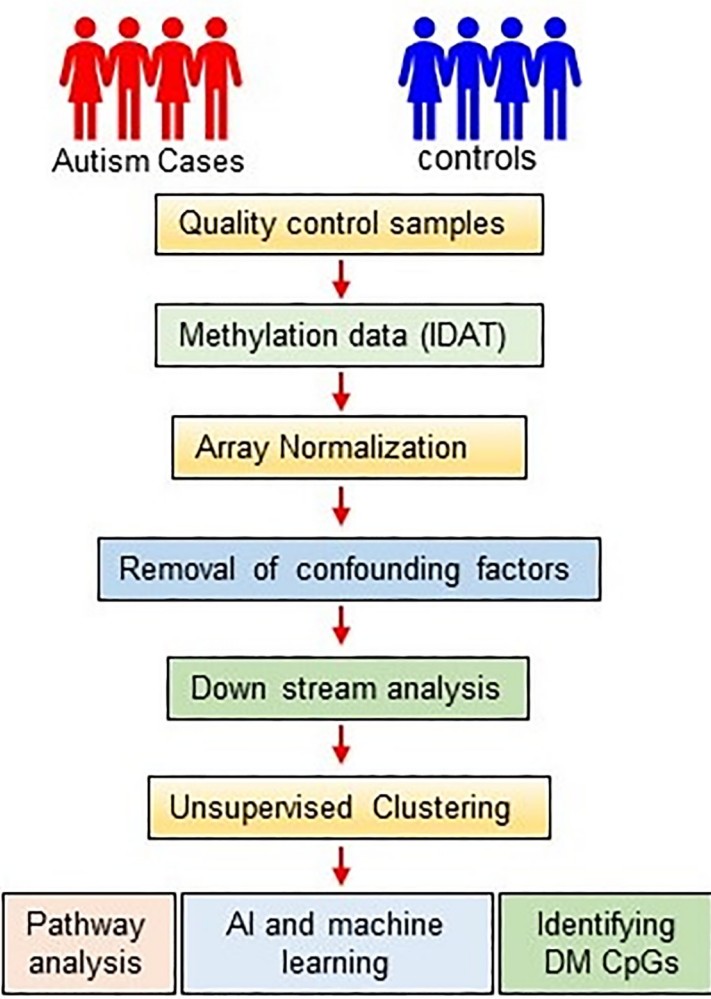

**Fig 1. Schematic summary of the methylation array data processing and analysis pipeline.**

For intragenic CpG markers, we identified a total of 3820 CpGs were hypo and 3033 were hypermethylated. The top 5 differentially methylated CpGs based on the lowest FDR p-values were: cg16699528 (*GATS*; *PVRIG*), cg15436096 and cg21893185 (*GPR135*), cg19949776 (*LOC100132724*; *AP4E1*) and cg13342370 (*ITGBL1*). All 5 of these CpGs displayed a methylation difference of ≥30% and were hypermethylated. The S2 Table provides the details of all significant CpG loci in autism full-term placental tissue.

## AI analysis results

AI analysis using combined inter-and intragenic markers was highly accurate for term autism detection. Based on DL analysis: AUC (95% CI) = 1.00 (1.00–1.00) with sensitivity and specificity of 100%. RF achieved identical performance. Analysis performed combining both inter- and intragenic markers also achieved AUC (95% CI) = 1.00 (1.00–1.00). The top 5 predictive markers were, cg23920016 (*NOS1AP*), cg24274662 (*MOSPD1*), cg05036212 (intergenic), cg26017408 (*AFAP1L2*) and cg16930349 (*GRIPAP1*) with DL and cg22914188 (*ANAPC7*), cg21483475 (*DHX36*), cg00991994 (*C3orf26*; *FILIP1L*; *MIR548G*), cg10706649 (*POLA2*) and

**Table 1. Results of Term Autism Placenta based on combined Inter and intragenic markers (with FDR p-value <0.05).**

|  | SVM | GLM | PAM | RF | LDA | DL |
|---|---|---|---|---|---|---|
| AUC | 0.9998 | 0.9988 | 0.9997 | 1.0000 | 0.9978 | 1.0000 |
| 95% CI | (0.9500–1) | (0.9500–1) | (0.9500–1) | (0.9900–1) | (0.9500–1) | (1–1) |
| Sensitivity | 0.9300 | 0.9000 | 1.0000 | 1.0000 | 0.9000 | 1.0000 |
| Specificity | 0.9200 | 0.9900 | 1.0000 | 1.0000 | 0.9500 | 1.0000 |

cg09856604 (intergenic) using RF AI platforms, Table 1. The evaluation based on AI using intragenic and intergenic separately are presented in S3A Table and S3B Table respectively.

**Important predictors in order**

SVM: cg25650964, cg23082393, cg18675381, cg21964564, cg23925650

GLM: cg24506662, cg19711553, cg16449972, cg18675381, cg17146731

PAM: cg18675381, cg01812571, cg21964564, cg03582285, cg02478023

RF  : cg22914188, cg21483475, cg00991994, cg10706649, cg09856604

LDA: cg11263351, cg18536607, cg13687570, cg11123972, cg24506662

DL  : cg23920016, cg24274662, cg05036212, cg26017408, cg16930349

## Biological functional enrichment analysis

The biological functional enrichment showed four biological functions to be significantly over-represented. The four functions are: (i) Quantity of synapse (p-6.37E-19), (ii) Microtubule dynamics (p-6.06E-8), (iii) Neuritogenesis (p-1.68E-7) and (iv) Abnormal morphology of neurons (p-5.99E-7) (Fig 2). Among the enriched genes in the above said biological functions, about 93% were hypomethylated and 7% were hypermethylated genes. These molecular pathways are relevant to neuronal dynamics, cognition, and autism. The relevance of these biological functions is further discussed.

## Discussion

The prevalence of ASD continues to increase in the US. Despite advances in our understanding of its biology the disease mechanisms remain incompletely understood. Currently, the detection of ASD rests on clinical and parental observations of childhood behavior [25]. The resulting delayed diagnosis contributes to delayed interventions [26] and worse outcomes. The placenta has the potential to act as a surrogate tissue to predict ASD and is characterized by methylated markers [15]. Based on the clear need to further understand disease mechanisms and for biological disease markers, we investigated the value of AI analysis of placental epigenomics for autism prediction. High diagnostic accuracy with AUC close to 1.0 was achieved with each of the six AI platforms for the detection of autism using placental epigenomics. This was observed when intragenic, intergenic (outside of known genes) CpGs were analyzed separately or in combination.

While preterm births are at higher risk for autism most cases occur in term births, by far the largest delivery group. While there is an overall male predominant of males in autism this is primarily due to their higher frequency among the term birth group with relatively higher female frequency among the preterm births [27]. This suggests that the mature male brain might be more susceptible to insults leaning to autism or that at the very least different mechanisms for ASD exists in term from preterm births. Based on this reasoning, we focused on the term autism in this study.

We further investigated the mechanisms of term autism. The prenatal environment can significantly affect neurodevelopment and appears to play a significant role in the etiology of

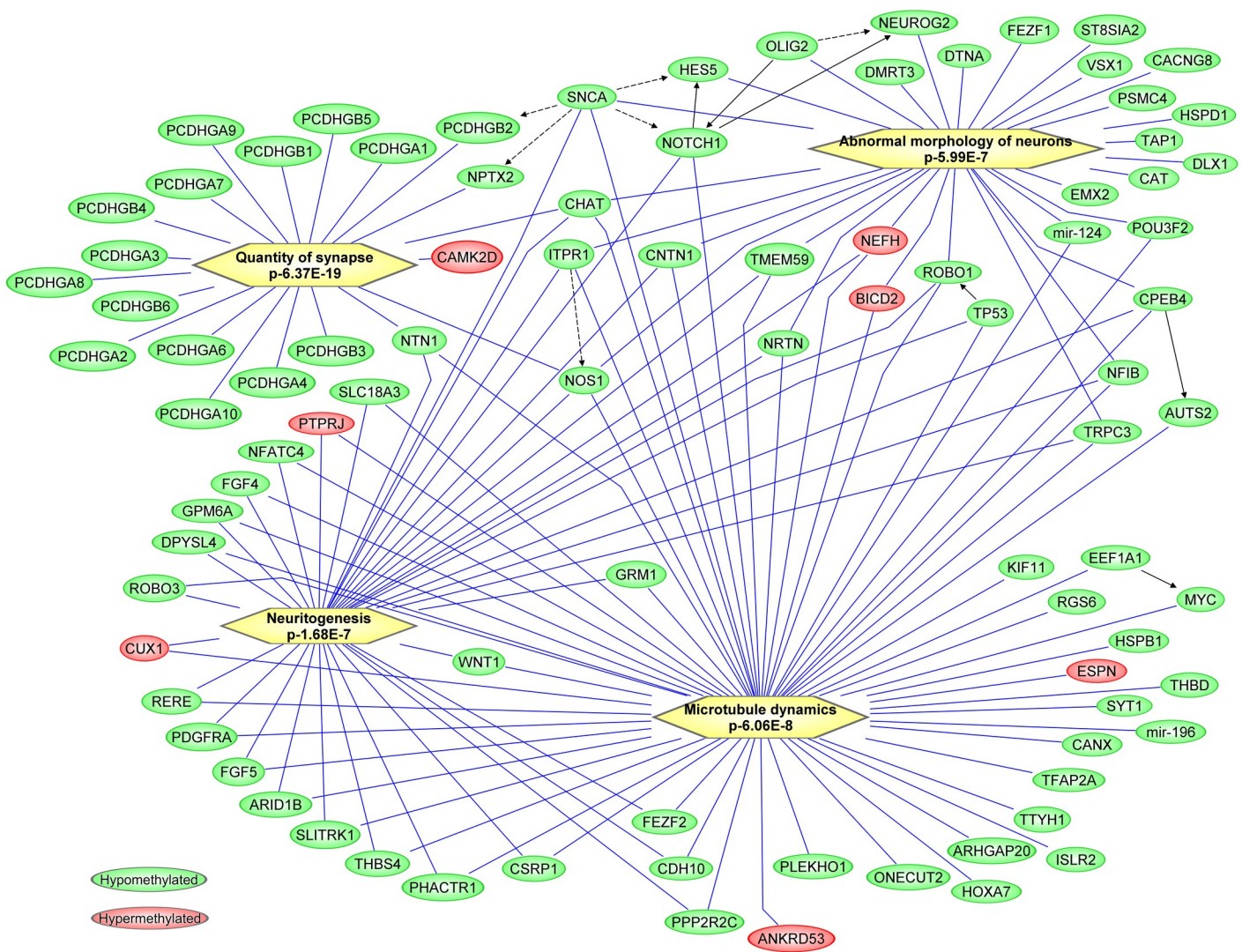

**Fig 2. Ingenuity Pathway Analysis pathways analysis of significant DNA methylation variations and network analysis performed using Ingenuity Pathway Analysis (IPA).**

ASD [28]. The placenta is the sole organ for the transmission of nutrients to the developing fetus that is required for brain development as well [9]. One study, using the placenta, identified two genes in association with ASD (*CYP2E1* and *IRS2*) [29] and we identified both to be differentially methylated as well.

Four genes cg23920016 (*NOS1AP*), cg24274662 (*MOSPD1*), cg26017408 (*AFAP1L2*) and cg16930349 (*GRIPAP1*) identified based on predictive ability for autism using AI analysis (Table 1). The *NOS1AP* (Neuronal Nitric Oxide Synthase 1 Adaptor Protein) gene is located at 1q23.3 which is a candidate region for several psychiatric conditions including ASD. Genetic variants of the *NOS1AP* gene are associated with ASD and other psychiatric diseases [30, 31]. The *MOSPD1* "Motile Sperm Domain Containing 1" gene contains the Major sperm protein (MSP) domain. MSP domain is a large family of genes whose proteins are involved in synaptic transmission in the central nervous system identified in sea slug and has a role in human neurodegenerative disorders [32]. *MOSPD1* is involved in Mesenchymal stem cells (MSCs) proliferation and differentiation. The MSCs exhibits immunomodulatory properties that

mediate diseases allied to inflammation and tissue damage such as ASD [32, 33]. The specific relationship between this gene and the term autism which is predominantly a male disorder remains to be elucidated. *AFAP1L2* (another gene identified based on AI analysis) showed differential expression in oligodendrocytes of the Amyotrophic lateral sclerosis murine model [34]. However, molecular studies are required to explore its role in ASD. *GRIPAP1* is one of the glutamate receptor interactors, involved in neuronal cytoskeleton organization [35, 36]. *GRIPAP1* genetic variant that alters the splicing of this gene was identified in ASD cases [37].

We further used the approach of IPA analysis to investigate the mechanisms of term autism. Significant over-representation of biological functions found were as follows: Quantity of synapse, Microtubule dynamics, Neuritogenesis and Abnormal morphology of neurons to be enriched with the significantly differentially methylated genes. The nature and potential role of each pathway are briefly discussed below.

## Quantity of synapse

A process of synapse formation occurs between neurons is required for communication. Synapse formation occurs in the fetus before 27 weeks gestation [38]. The synaptic connections established by the neurons integrate functional neuronal networks required for the appropriate brain function. Disruption of this function is associated with ASD pathologies including cognitive impairments [39]. Some constituent genes in this pathway that were found to be epigenetically altered in our study are highlighted. Protocadherin (Pcdh) is a family of genes involved in the formation of neural circuits and synaptogenesis. Multiple studies implicate gene mutations, copy number variations, and epigenetic variations in the Pcdh family of genes in neurodevelopment and neurological diseases [40]. The present study found multiple PCDH family genes to be hypomethylated and on enrichment analysis, PCDH family genes are enriched with "quantity of synapse" (Fig 2). CAMK2D, a hypermethylated gene has a role in intracellular calcium signaling and has been associated with ASD [41].

## Microtubule dynamics

Neurons rely on microtubule cytoskeleton dynamics for several processes such as cell division, cell migration, intracellular trafficking, signal transduction, axon guidance, and synapse formation. After synapse formation, microtubules provide physical integrity that maintains neural connectivity throughout the developmental process [42]. Microtubule-associated proteins support the microtubular functions that help axon outgrowth and pathfinding as well as dendrite development [43]. Alterations in the levels of microtubule-associated proteins have been identified in ASD patients [43]. The gene MYC was found to be differentially methylated in the current study and was previously reported to be differentially expressed in neurological disorders [44]. Upregulation of gene activity was identified in ASD cases [45]. The hypomethylated *WNT1* gene was earlier found to be associated with a missense polymorphism in ASD and has a probable role in inducing Wnt signaling pathway activation [46]. We identified altered methylation on other WNT family genes, *WNT2*, *WNT2B*, *WNT7B*, and *WNT10A* in the present study.

## Neuritogenesis

Neuritogenesis is one of the primary events during neuronal development in which new neurites or growth cones form and give rise to axons and dendrites [47]. The control of neuritogenesis is complex and evident in neuronal connectivity deficits in ASD and mutations in the genes affecting neuritogenesis have been associated with autism [48]. One of the autism candidate genes, AUTS2 is involved in neuritogenesis via Rac1 signaling activation in the

developing brain [49]. The *CUX1* gene in the neuritogenesis pathway was found to be hyper-methylated in our study. Mutations in the active enhancer regions were associated with social behavior and cognitive function of ASD in an earlier study by Doan et al., 2016 [50].

## Abnormal morphology of neurons

Alterations in the neuronal cytoskeleton composition, especially aberrations in the actin fila-ment polymerization, have been correlated with ASD development [51]. In the postmortem human brains with ASD, both cortical and non-cortical regions showed abnormal neuronal morphology, suggesting the importance of neuronal morphology in the pathogenesis of ASD [52]. *NEUROG2* (*NGN2*), found to be hypomethylated in our study, is a transcription factor that converts progenitors to a neuronal fate during brain development. This also reprograms early postnatal astroglia to develop into neurons [53]. *NGN2* can induce excitatory neurons in human cortices, and *NGN2* knockout cells lack these neurons, indicating the *NGN2* may be a key gene in ASD [54, 55]. Similarly, another of the placental hypomethylated genes, *POU3F2*, is a transcription factor that contributes to the process of neuronal differentiation [56]. This gene is highly expressed in the developing brain and is said to be a master regulator of down-stream ASD candidate genes [57]. Given its role as a fetal tissue and the biological regulator of physiological and pathological interactions between mother and fetus, it is not surprising that the placenta is considered as an appropriate surrogate for the evaluation of a fatal brain disor-der. Prior studies have shown changes in the placental morphology [28] suggestive of reduced ability to respond to intrauterine stresses. In addition, reduced placental trophoblast branch-ing, the key vascular unit for exchange of materials and waste products between mother and fetus was one of the changes reported. Differential methylated regions (DMRs) of the placenta DNA have been reported to distinguish ASD cases and controls [29]. These DMRs were func-tionally enriched for neuronal development. The findings reached genome-wide significance for two genes: *CYP2E1* and *IRS2*. Similarly, methylation differences were found in the placen-tal methylomes in both partially and highly methylated regions of the DNA in ASD placentas compared to unaffected controls [15]. This reached genome-wide significance near the *DLL1* gene, which is thought to be a potential fetal brain enhancer. Collectively, these studies support our findings that placental epigenomic alterations are a feature of ASD. Our study however uses single nucleotide level resolution to improve specificity to both investigate the pathogene-sis of term autism, accurately predict term autism. The identified key genes and pathways involved in ASD pathogenesis provide an opportunity to identify novel targets that can be uti-lized for therapeutic development.

Our study suffers from the limitations of a small sample size and the lack of corresponding gene expression data to further refine the potential biological consequences of the epigenetic changes. While expression studies are possible from FFPE samples, in our hands, they have not performed as well as fresh placental tissues. It is also possible that co-existing obstetric con-ditions such as fetal growth restriction and preeclampsia can modify the placental methylome. However, we excluded cases with significant obstetric or other complications. Further, using AI and pathway analysis we demonstrate that the most significant genes and pathways appear highly relevant to brain and neuronal development, providing greater biological plausibility for our results.

## Conclusions

Combining DNA profiling with AI analysis we achieved accurate early prediction of term autism cases. Given the importance of early diagnosis and intervention for improved outcomes in ASD, our findings could be clinically significant. Biologically relevant functions of identified

genes include synaptic transmission, neuronal cytoskeleton organization, neuritogenesis, the morphology of neurons which are pertinent to ASD. We used a limited number of samples and further functional analysis is desirable. Our findings should therefore be preliminary at this stage and of need for validation in larger data sets. However, the study confirms the potential utility of the placenta as a surrogate tissue, given its easy accessibility and our findings of high accuracy.

## Supporting information

**S1 Table. Clinical and demographic characteristics: Full-term Autism cases versus control subjects.**
(DOCX)

**S2 Table. Details of all significant CpG loci in autism full-term placenta tissue.** Target ID, Gene ID, chromosome, p-value, FDR p-values, % methylation, and AUC ROC details are given.
(XLSX)

**S3 Table.** A. Results of Term Autism Placenta based on intragenic CpGs only (with FDR p-value < 0.05). B. Results of Term Autism Placenta based on intergenic (nongenic) markers only (with FDR p-value < 0.05).
(DOCX)

**S1 Fig. Principal Component Analysis (PCA) showed clear separation of ASD cases and normal control subjects.**
(TIF)

**S2 Fig. Heat map showing methylation variation in autism cases compared with controls.** The hierarchical clustering showed separation of CpG markers based on hyper and hypo-methylation status depicted on the figure completely methylated (red) to unmethylated (blue).
(TIF)

## Acknowledgments

We thank the Michigan Department of Health and Human Services for specimens.

## Author Contributions

**Conceptualization:** Ray O. Bahado-Singh, Uppala Radhakrishna.

**Data curation:** Ray O. Bahado-Singh, Sangeetha Vishweswaraiah, Buket Aydas, Uppala Radhakrishna.

**Formal analysis:** Sangeetha Vishweswaraiah, Buket Aydas, Uppala Radhakrishna.

**Funding acquisition:** Ray O. Bahado-Singh.

**Investigation:** Ray O. Bahado-Singh, Uppala Radhakrishna.

**Methodology:** Sangeetha Vishweswaraiah, Buket Aydas, Uppala Radhakrishna.

**Project administration:** Ray O. Bahado-Singh, Uppala Radhakrishna.

**Resources:** Ray O. Bahado-Singh, Uppala Radhakrishna.

**Software:** Sangeetha Vishweswaraiah, Buket Aydas, Uppala Radhakrishna.

**Supervision:** Ray O. Bahado-Singh, Buket Aydas, Uppala Radhakrishna.

**Validation:** Sangeetha Vishweswaraiah, Buket Aydas, Uppala Radhakrishna.

**Visualization:** Uppala Radhakrishna.

**Writing – original draft:** Ray O. Bahado-Singh, Sangeetha Vishweswaraiah, Buket Aydas, Uppala Radhakrishna.

**Writing – review & editing:** Ray O. Bahado-Singh, Sangeetha Vishweswaraiah, Buket Aydas, Uppala Radhakrishna.

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
