## [Decision Letter · Decision Letter 0]

26 Mar 2021

PONE-D-21-03311

Placental DNA methylation changes and the early prediction of autism in full-term newborns

PLOS ONE

Dear Dr. Radhakrishna,

Thank you for submitting your manuscript to PLOS ONE. After careful consideration, we feel that it has merit but does not fully meet PLOS ONE’s publication criteria as it currently stands. Therefore, we invite you to submit a revised version of the manuscript that addresses the points raised during the review process.

We look forward to receiving your revised manuscript.

Kind regards,

Abhishek Kumar, Ph.D.

Academic Editor

PLOS ONE

Journal Requirements:

3. We note that you are reporting an analysis of a microarray, next-generation sequencing, or deep sequencing data set. PLOS requires that authors comply with field-specific standards for preparation, recording, and deposition of data in repositories appropriate to their field. Please upload these data to a stable, public repository (such as ArrayExpress, Gene Expression Omnibus (GEO), DNA Data Bank of Japan (DDBJ), NCBI GenBank, NCBI Sequence Read Archive, or EMBL Nucleotide Sequence Database (ENA)). In your revised cover letter, please provide the relevant accession numbers that may be used to access these data. For a full list of recommended repositories, see http://journals.plos.org/plosone/s/data-availability#loc-omics or http://journals.plos.org/plosone/s/data-availability#loc-sequencing.

4.  Thank you for stating the following in the Financial Disclosure section:

We note that one or more of the authors are employed by a commercial company: "Department of Healthcare Analytics,"

Additional Editor Comments (if provided):

Firstly, I am sorry that for delays but it was required for reviewing process.

Please provide answers to question raised by the reviewer. I want to add a few points for the improvement of this manuscript.

a) Please provide a pipeline figure for methods followed in this study.

b) Provide details of the pathway data in the result section and roles of these four pathways in autism.

Looking forward for these changes.

Sincerely

Dr. Abhishek Kumar

Reviewers' comments:

Reviewer's Responses to Questions

**Comments to the Author**

1. Is the manuscript technically sound, and do the data support the conclusions?

Reviewer #1: Yes

2. Has the statistical analysis been performed appropriately and rigorously? 

Reviewer #1: Yes

3. Have the authors made all data underlying the findings in their manuscript fully available?

Reviewer #1: Yes

4. Is the manuscript presented in an intelligible fashion and written in standard English?

Reviewer #1: Yes

5. Review Comments to the Author

Reviewer #1: Current Manuscript describes the use of placental DNA methylation alterations and the early prediction of autism in full-term newborns. Autism spectrum disorder (ASD) have been previously linked with abnormal brain development and epigenetic dysfunctions during early development stages. The authors hypothesized if placental DNA methylation changes could be used as a prediction tools to elucidate the early pathogenesis of ASD. They performed genome-wide methylation analysis of 14 patients and 10 controls using placental tissue followed Ingenuity Pathway Analysis, six Artificial Intelligence (AI) algorithms including Deep Learning (DL) to investigate the predictive accuracy of CpG markers for autism detection. They identified important biological pathways and genes involved early fetal neurodevelopmental process that influence later cognition and social behavior. Overall, this is an excellent study that provides important insights about the biological pathways involved in ASD. The authors should address the following minor concerns in order to improve the quality of their manuscript.

1. The authors performed genome-wide methylation analysis using 14 patients and 10 controls. The gender of these individuals is not clear. The authors should clearly specify this in the table and methods section.

2. There are few incomplete statements in the manuscript. Line 239, they stated “Further functional studies are required to explore its role in ASD”. What do you mean by further functional analysis here? Little more clarity would be helpful.

3. The authors should consider including few statements in the discussion section if the biological pathways identified in this study are druggable.

4. There are multiple typos, grammatical and syntax errors that the authors should correct.

6. PLOS authors have the option to publish the peer review history of their article (what does this mean?). If published, this will include your full peer review and any attached files.

Reviewer #1: No

---

## [Author Response · Author response to Decision Letter 0]

21 May 2021

Thank you for the opportunity to revise and resubmit our manuscript. We sincerely appreciate all your valuable comments and suggestions which have helped us improve the quality of the article. We believe that the revisions based on these comments have made this a better and clearer paper.

Additional Editor Comments:

Please provide answers to question raised by the reviewer. I want to add a few points for the improvement of this manuscript.

a) Please provide a pipeline figure for methods followed in this study.

Author's Response: Thank you for your comment. We have included the Schematics of the study design in the methodology section as fig 1.

b) Provide details of the pathway data in the result section and roles of these four pathways in autism.

Author's response: We have specified the biological process and provided the significance in terms of p-value in the result section. Added the details of hypo- and hyper-methylated genes and have discussed their relevance to autism in the discussion section. 

Result section: The biological functional enrichment showed four biological functions to be significantly overrepresented. The four functions are: (i) Quantity of synapse (p-6.37E-19), (ii) Microtubule dynamics (p-6.06E-8), (iii) Neuritogenesis (p-1.68E-7) and (iv) Abnormal morphology of neurons (p-5.99E-7) (Fig 1). Among the enriched genes in the above said biological functions, about 93% were hypomethylated and 7% were hypermethylated genes. These molecular pathways are relevant to neuronal dynamics, cognition, and autism. The relevance of these biological functions is further discussed.

5. Review Comments to the Author

Reviewer #1: Current Manuscript describes the use of placental DNA methylation alterations and the early prediction of autism in full-term newborns. Autism spectrum disorder (ASD) has been previously linked with abnormal brain development and epigenetic dysfunctions during the early development stages. The authors hypothesized if placental DNA methylation changes could be used as prediction tools to elucidate the early pathogenesis of ASD. They performed genome-wide methylation analysis of 14 patients and 10 controls using placental tissue followed Ingenuity Pathway Analysis, six Artificial Intelligence (AI) algorithms including Deep Learning (DL) to investigate the predictive accuracy of CpG markers for autism detection. They identified important biological pathways and genes involved in the early fetal neurodevelopmental process that influence later cognition and social behavior. Overall, this is an excellent study that provides important insights about the biological pathways involved in ASD. The authors should address the following minor concerns to improve the quality of their manuscript.

1. The authors performed genome-wide methylation analysis using 14 patients and 10 controls. The gender of these individuals is not clear. The authors should specify this in the table and methods section.

Author's Response: Thank you for your advice, now we have included the details in the methods section. 

2. There are few incomplete statements in the manuscript. Line 239, stated, “Further functional studies are required to explore its role in ASD”. What do you mean by further functional analysis here? A little more clarity would be helpful.

Author's response: The statement has been modified “However, molecular studies are required to explore its role in ASD.”

3. The authors should consider including few statements in the discussion section if the biological pathways identified in this study are druggable.

Author's response: We have included the statement in the discussion section (line 324-326)

4. There are multiple typos, grammatical and syntax errors that the authors should correct.

Author's response: We have made language corrections in the manuscript. (track changes).

---

## [Editor Report · Decision Letter 1]

3 Jun 2021

Placental DNA methylation changes and the early prediction of autism in full-term newborns

PONE-D-21-03311R1

Dear Dr. Radhakrishna,

We’re pleased to inform you that your manuscript has been judged scientifically suitable for publication and will be formally accepted for publication once it meets all outstanding technical requirements.

Kind regards,

Abhishek Kumar, Ph.D.

Academic Editor

PLOS ONE

Additional Editor Comments (optional):

We are glad to inform that this manuscript is accepted for publication.
---

## [Editor Report · Acceptance letter]

15 Jun 2021

PONE-D-21-03311R1 

Placental DNA methylation changes and the early prediction of autism in full-term newborns 

Dear Dr. Radhakrishna:

I'm pleased to inform you that your manuscript has been deemed suitable for publication in PLOS ONE. Congratulations! Your manuscript is now with our production department. 

Kind regards, 

on behalf of

Dr. Abhishek Kumar 

Academic Editor

PLOS ONE